# Structural variation, functional differentiation and expression characteristics of the AP2/ERF gene family and its response to cold stress and methyl jasmonate in *Panax ginseng* C.A. Meyer

**Jing Chen**[1,2◉], **Yuanhang Zhou**[1◉], **Qi Zhang**[1], **Qian Liu**[1], **Li Li**[1], **Chunyu Sun**[1,2], **Kangyu Wang**[1,2], **Yanfang Wang**[2,3], **Mingzhu Zhao**[1,2], **Hongjie Li**[1], **Yilai Han**[1], **Ping Chen**[1], **Ruiqi Li**[1], **Jun Lei**[1], **Meiping Zhang**[1,2]*, **Yi Wang**[1,2]*

1 College of Life Science, Jilin Agricultural University, Changchun, Jilin, China, 2 Research Center for Ginseng Genetic Resources Development and Utilization, Changchun, Jilin, China, 3 College of Chinese Medicinal Materials, Jilin Agricultural University, Changchun, Jilin, China

◉ These authors contributed equally to this work.
* meiping.zhang@jlau.edu.cn (MZ); wanglaoshi0606@163.com (YW)

**Data Availability Statement:** All relevant data are within the paper and its Supporting Information files.

## Abstract

The APETALA2/Ethylene Responsive Factor (AP2/ERF) gene family has been shown to play a crucial role in plant growth and development, stress responses and secondary metabolite biosynthesis. Nevertheless, little is known about the gene family in ginseng (*Panax ginseng* C.A. Meyer), an important medicinal herb in Asia and North America. Here, we report the systematic analysis of the gene family in ginseng using several transcriptomic databases. A total of 189 putative *AP2/ERF* genes, defined as *PgERF001* through *PgERF189*, were identified and these *PgERF* genes were spliced into 397 transcripts. The 93 *PgERF* genes that have complete AP2 domains in open reading frame were classified into five subfamilies, DREB, ERF, AP2, RAV and Soloist. The DREB subfamily and ERF subfamily were further clustered into four and six groups, respectively, compared to the 12 groups of these subfamilies found in *Arabidopsis thaliana*. Gene ontology categorized these 397 transcripts of the 189 *PgERF* genes into eight functional subcategories, suggesting their functional differentiation, and they have been especially enriched for the subcategory of nucleic acid binding transcription factor activity. The expression activity and networks of the 397 *PgERF* transcripts have substantially diversified across tissues, developmental stages and genotypes. The expressions of the *PgERF* genes also significantly varied, when ginseng was subjected to cold stress, as tested using six *PgERF* genes, *PgERF073*, *PgERF079*, *PgERF110*, *PgERF115*, *PgERF120* and *PgERF128*, randomly selected from the DREB subfamily. This result suggests that the DREB subfamily genes play an important role in plant response to cold stress. Finally, we studied the responses of the *PgERF* genes to methyl jasmonate (MeJA). We found that 288 (72.5%) of the 397 *PgERF* gene transcripts responded to the MeJA treatment, with 136 up-regulated and 152 down-regulated, indicating that most members of the *PgERF* gene family are responsive to MeJA. These results,

**Funding:** This research was supported by an award from China 863 Project (2013AA102604-3), the Bureau of Science and Technology of Jilin Province (20190201264JC, 20170101010JC, 20180414077GH, 20180101027JC), the Development and Reform Commission of Jilin Province (2016C064, 2018C047-3), and a startup fund from Jilin Agriculture University (201801, https://www.jlau.edu.cn/).

**Competing interests:** The authors have declared that no competing interests exist.

therefore, provide new resources and knowledge necessary for family-wide functional analysis of the *PgERF* genes in ginseng and related species.

## Introduction

Plants are subjected to numerous biotic and abiotic stresses all time through their growth and development. Therefore, they have developed a variety of mechanisms by producing secondary signaling molecules (e.g., ethylene and jasmonic acid) and responsive networks at the molecular, biochemical and physiological levels to perceive the external signals from the stresses and to response to the stresses [1]. It has been documented that a large number of genes are involved in these processes [2]. Therefore, it is important to decipher the regulatory mechanisms of the defense-related genes involved in the signal transduction pathways and the plant responses to these stresses for enhanced plant genetic improvement [3]. The APETALA2/Ethylene Responsive Factor (AP2/ERF) transcription factors have been demonstrated to be one of the most important gene families actively functioning in plant response to biotic and abiotic stresses by binding to cis-acting elements of downstream target genes [4].

The AP2/ERF family has one or two conserved APETALA2 (AP2) domains having 60–70 amino acids [5]. Based on the number and amino acid sequence similarities of the AP2 domains, the AP2/ERF family is divided into the DREB (dehydration responsive element binding), ERF, AP2, RAV (Related to ABI3/VP1) and Soloist subfamilies [6,7]. Both DREB and ERF subfamilies possess a single AP2 domain, with a specific WLG motif, and could be subdivided into A1 to A6 and B1 to B6 groups, respectively [6]. Alternatively, the DREB and ERF subfamilies were categorized into I to VI, and V to X, VI-L and Xb-L groups, respectively [7]. The AP2 subfamily has two tandemly repeated AP2 domains, while the RAV subfamily has one AP2 domain and one B3 domain that are commonly found in other transcription factors [8]. The Soloist subfamily also has only one AP2 domain. It was classified into an independent subfamily due to its relatively low sequence homology with the DREB and ERF subfamilies [9]. Although the AP2 domain of the AP2/ERF family is highly conserved, its five subfamilies, DREB, ERF, AP2, RAV and Soloist, recognize different DNA cis-acting elements and exhibit substantial functional diversity [10]. Specifically, the members of the AP2 subfamily bind to the GCAC(A/G)N(A/T)TCCC(A/G)ANG(C/T) element and regulate developmental processes of different plant tissues, e.g., embryo, flower, sepal and fruit [11–13]. The *RAV1* gene of the RAV subfamily was reported to bind to CAACA and CACCTG motifs in *Arabidopsis thaliana* (Arabidopsis) [14]. The roles of the RAV subfamily in plant development and responses to various biotic and abiotic stresses were investigated in several plant species [15–17]. The only gene of the Soloist subfamily in Arabidopsis, *APD1* (*At4g13040*), worked as a positive regulator of disease defense by up-regulating the accumulation of salicylic acid (SA) [18]. The members of the ERF subfamily typically bind to the cis-acting element GCC-box and are involved in the signaling pathways of plant hormone, e.g., ethylene (ET), SA, jasmonic acid (JA) and abscisic acid (ABA), and play an important role in both plant growth and development and response to stresses [19–21]. On the other hand, the DREB subfamily recognizes the conserved CCGAC motif of the dehydration-responsive element present in stress-responsive genes and is associated with plant response to abiotic stresses [22–24].

The AP2/ERF family has been well characterized in the model plant species, Arabidopsis [6,7] and *Medicago truncatula* [25], and several crops, such as rice [7], maize [26], soybean [27], Chinese cabbage [10], grapevine [28] and *Populus trichocarpa* [29]. However, little is

known about the AP2/ERF family in the medicinal herb, ginseng (*Panax ginseng* C.A. Meyer). Ginseng is a perennial of the Araliaceae family and has long been cultivated for human medicine in North America and Asia, particularly in China, Korea, and Japan. Ginseng, known as the "king of all herbs" in China, is mainly cultivated in Jilin Province; therefore, it is often known as Jilin ginseng. Ginseng has been widely used as a medicinal herb due to its bioactive components, especially ginsenosides that have been shown to play significant roles in anti-inflammation [30,31], anti-tumor [32], and immunomodulation [33]. However, ginseng has been suffering from various biotic and abiotic stresses, which is greatly threating its production. Therefore, identification, characterization and utilization of the defense-related genes in ginseng are of significance for ginseng breeding and production. In the present study, we comprehensively studied the AP2/ERF family in Jilin ginseng in several aspects, including gene identification, protein motif characterization, functional categorization and phylogenetic analysis. Moreover, the expression activities and patterns of AP2/ERF genes were also investigated at different developmental stages, in different tissues, different cultivars, under cold stress and under the methyl jasmonate (MeJA) treatment. The results of these studies have laid the foundation for deep functional analysis and utilization of the genes of the AP2/ERF family and provided vital information on the molecular mechanism of plant response to biotic and abiotic stresses in ginseng and related plant species.

## Materials and methods

### Databases

We previously established a comprehensive transcriptome for Jilin ginseng from 14 tissues (fiber root, leg root, main root epiderm, main root cortex, rhizome, arm root, stem, leaf peduncle, leaflet pedicel, leaf blade, fruit peduncle, fruit pedicel, fruit flesh, and seed), from which 248,993 transcript unigenes (130,557 gene IDs) were assembled [34]. Moreover, we also sequenced and established the databases for the root transcriptomes sampled from 5-, 12-, 18- and 25-year-old plants [34] and four-year-old plants of 42 genotypes (named from S1 to S42) representing the diversity of Jilin ginseng [35]. In this study, a ginseng line IR826 genome sequence database [36] and a Ginseng Genome Database (http://ginsengdb.snu.ac.kr/index. php) [37] were also used. In addition, a transcriptome database of the adventitious roots of ginseng cv. Cheongsun treated with 200 μM MeJA for 0, 12, 24 and 48 h, respectively [38], was consulted. These databases all could be retrieved through their publications as indicated above.

### Identification of *PgERF* genes in ginseng

To identify the genes of the AP2/ERF family in ginseng, the Hidden Markov Model (HMM) profile of the AP2/ERF domain (Pfam: PF00847) and the protein sequences of the AP2/ERF genes downloaded from NCBI (http://blast.ncbi.nlm.nih.gov/Blast) were used to query the 248,993 Jilin ginseng transcript unigenes [34] by TBLASTN at E-value ≤ le-6. The obtained sequences were then used as queries to search for homologs in the ginseng line IR826 genome database [36]. Furthermore, TBLASTN was performed again to search the 248,993 transcript unigenes [34] using the homologs as queries with E-value ≤ le-6 to maximize identification of the AP2/ERF genes in ginseng. The identified AP2/ERF gene homologues were defined as *PgERF* for the AP2/ERF genes in ginseng and extracted by a Perl programming software. Finally, the predicted *PgERF* genes were analyzed by the conserved domain database (CDD) (http://www.ncbi.nlm.nih.gov/Structure/cdd/wrpsb.cgi) and the ORF Finder (http://www.ncbi.nlm.nih.gov/gorf/gorf.html) at NCBI.

## Multiple sequence alignment and phylogenetic analysis of *PgERF* genes

The 93 *PgERF* genes that have complete AP2 domains have been selected and their AP2 domains were aligned using the ClustalW program [39]. A phylogenetic tree was constructed from the 93 *PgERF* genes that represent the AP2/ERF family in ginseng with 147 *AtERF* genes previously identified and annotated in Arabidopsis [7]. The phylogenetic tree was constructed using MEGA 5.0 by the Neighbor-Joining (NJ) and the Maximum Likelihood (ML) methods, respectively, with 1,000 bootstrap replications, using the Poisson correction model and the pairwise deletion [40].

## Motif prediction of *PgERF* genes

The putative protein sequences of the 93 *PgERF* genes used for construction of the AP2/ERF family phylogenetic tree were subjected to the online software, MEME (multiple EM for motif elicitation, V5.0.3) (http://meme-suite.org/tools/meme) [41] to identify their conserved motifs. Motif length was set to 6–50 amino acids and the maximum number of motifs was set to 25, while other parameters were set as default.

## Expression and functional networks of *PgERF* genes

The expression profiles of all putative transcripts of *PgERF* genes identified above were extracted by a Perl programming software from the above four transcriptome databases: (1) the 14 tissues of a four-year-old Jilin ginseng plant, (2) the roots of Jilin ginseng 5-, 12-, 18- and 25-year-old plants, (3) the four-year-old roots of 42 Jilin ginseng genotypes and (4) the ginseng cv. Cheongsun adventitious roots treated with 200 μM MeJA for 0, 12, 24 and 48 h, respectively. The expression profiles of the putative transcripts of *PgERF* genes were measured as transcripts per million (TPM) and visualized by expression heatmap using the R programming language and software (http://www.r-project.org/, V3.3.3). Finally, the co-expression networks of these *PgERF* gene transcripts were constructed and analyzed among different tissues and different genotypes of Jilin ginseng using the BioLayout Express[3D] software (Version 3.2) [42]. The Pearson's correlation coefficients of expression between genes were used for the co-expression network construction at a two-tailed significance level of $p \leq 0.05$ or lower.

## Expression activity of *PgERF* genes responding to cold stress

One gram of equivalent ginseng hair roots were freshly cut from mature hair roots and cultured in 250 ml 1/2 Murashige and Skoog (MS) medium in dark at 22˚C for 30 days. Then, the 30-day-old hair root culture was subjected to cold stress at 4˚C for 0 h, 6 h, 24 h, 48 h and 72 h, respectively, with three replications per time point. Afterwards, the ginseng hair roots were harvested, frozen in liquid nitrogen and stored at -80˚C for RNA isolation and quantitative real-time PCR (qRT-PCR) analysis. Total RNA was extracted by TRIzol reagent (Bioteke, Beijing, China) and reverse-transcribed into cDNA using a PrimeScript™ RT reagent Kit with gDNA Eraser (TaKaRa, Tokyo, Japan). qRT-PCR was performed for six *PgERF* genes, *PgERF073*, *PgERF079*, *PgERF110*, *PgERF115*, *PgERF120* and *PgERF128*, that were randomly selected from the DREB subfamily. The *PgGADPH* gene was used as the internal reference. The gene-specific primers for qRT-PCR were designed by Primer Premier Software (version 5) (S1 Table). The qRT-PCR was conducted by Applied Biosystems 7500 Real Time PCR System (Thermo Fisher Scientific, Waltham, USA) and SYBR Premix Ex Taq™ II (TaKaRa, Tokyo, Japan). The qRT-PCR was performed using the following steps: 30 seconds at 95˚C; 40 cycles of 5 seconds at 95˚C and 34 seconds at 60˚C; one cycle of 15 seconds at 95˚C and 60 seconds at 60˚C; 15 seconds at 95˚C. All the experiments were amplified in triplicate. The relative

expression levels of these genes were calculated using formula $2^{-\Delta\Delta C}$ [43]. The expression levels of each gene were compared among different time points of the cold treatment by analysis of variance (ANOVA), followed by least significance difference (LSD).

## Results

### Identification and classification of *PgERF* genes

A total of 397 transcripts that were derived from 189 predicted *PgERF* genes, including those containing a partial or complete AP2/ERF domain, were identified. These *PgERF* genes were defined *PgERF001* to *PgERF189*, with a suffix (e.g., -1) for different transcripts derived from the same gene (S2 Table). Then, the sequences of 342 AP2/ERF gene transcripts downloaded from the Ginseng Genome Database (http://ginsengdb.snu.ac.kr/index.php) were aligned with 397 transcripts identified in this study with identity≥95%, alignment length≥200 bp (about AP2 maximum domain length). As a result, 302 (76%) of the 397 transcripts identified in this study that were spliced from 138 (73%) of the 189 *PgERF* genes were similar to 266 (78%) of the 342 transcripts of the AP2/ERF genes from the Ginseng Genome Database (S1 Fig and S3 Table). However, the remaining 51 (27%) *PgERF* genes whose sequences were quite different from the Korean ginseng AP2/ERF genes were considered as newly discovered AP2/ERF genes in ginseng (S1 Fig and S3 Table). The *PgERF* gene transcripts identified in this study had nucleotide sequences ranging from 203 bp to 2,897 bp, with an average length of 1,216 bp. Of the 397 *PgERF* gene transcripts, 176 that were spliced from 96 *PgERF* genes had partial AP2 domains or complete AP2 domains but being outside of ORFs. The remaining 221 *PgERF* transcripts, derived from 93 *PgERF* genes, had complete AP2 domains within ORFs. Therefore, these 221 *PgERF* transcripts were further analyzed. The 221 *PgERF* gene transcripts encode putative proteins with a length varying from 96 to 561 amino acids, with an average length of 256 amino acids (S4 Table). Analysis using the ExPASy Server showed that these putative proteins had an isoelectric point between 4.43 (*PgERF025*) and 11.12 (*PgERF180*) and a molecular mass ranging from 11.00 kDa (*PgERF140*) to 62.83 kDa (*PgERF159*) (S4 Table).

Sakuma et al. [6] classified the Arabidopsis AP2/ERF family into five subfamilies, ERF, DREB, AP2, RAV and Soloist. We also classified the 93 *PgERF* genes whose transcripts had complete AP2 domains within ORFs into these five subfamilies, according to the structures and the number of AP2/ERF domains. These five *PgERF* subfamilies, ERF, DREB, AP2, RAV and Soloist, contained 27, 48, 14, 2 and 2 genes, respectively (Fig 1). Specifically, two genes, *PgERF035* and *PgERF084*, were classified into the Soloist subfamily due to their low homology with the remaining AP2/ERF genes and their high homology with the Arabidopsis Soloist subfamily gene, *AT4G13040*. The *PgERF112* gene was the only one that codes one AP2/ERF domain with one B3 domain; therefore, it was classified into the RAV subfamily. Moreover, *PgERF171* was also classified into the RAV subfamily, even though it does not contain the B3 domain, because it has a high homology with the Arabidopsis RAV subfamily genes, *AT1G51120* and *AT1G50680*. As seven *PgERF* genes, *PgERF062*, *PgERF089*, *PgERF134*, *PgERF135*, *PgERF142*, *PgERF148* and *PgERF159*, contain two repeated AP2/ERF domains, they were classified into the AP2 subfamily. In addition, another seven *PgERF* genes, including *PgERF020*, *PgERF045*, *PgERF048*, *PgERF076*, *PgERF101*, *PgERF132* and *PgERF140*, were also classified into the AP2 subfamily as they have high sequence similarity with the members of the AP2 subfamily, even though they do not contain two repeated AP2/ERF domains. Of the 75 remaining *PgERF* genes, 48 and 27 were classified into the ERF and DREB subfamilies, respectively (Fig 1). Moreover, the DREB and ERF subfamilies of ginseng were each further divided into six groups, A1 through A6 and B1 through B6, respectively, or these two subfamilies were subdivided into four groups, from I to IV, and six groups, from V to X, respectively,

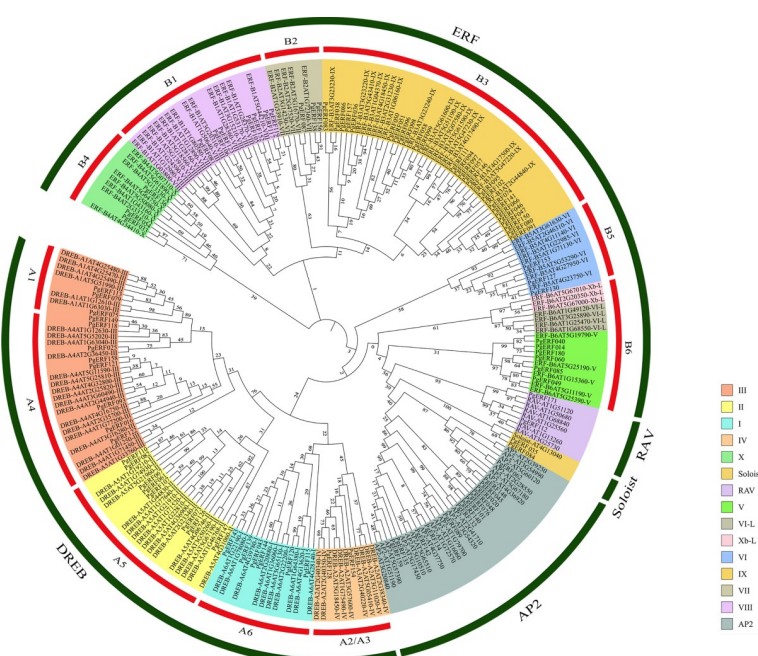

**Fig 1. Phylogenetic tree of the *AP2/ERF* gene family present in ginseng and Arabidopsis.** The amino acid sequences of the AP2 domain were aligned using Clustal W and the phylogenetic tree was constructed using neighbor-joining method.

but none of these genes were subdivided into the groups, VI-L and Xb-L, as were for the Arabidopsis AP2/ERF family, based on Nakano et al. [7] (Fig 1).

## Phylogenetic analysis of the *PgERF* gene family

To determine the phylogeny of the *PgERF* gene family, the 93 *PgERF* genes whose transcripts had complete AP2 domains were used. The phylogenetic tree was constructed with the NJ and ML methods, respectively. The NJ phylogenetic tree showed that the *PgERF* gene family was apparently classified into five clades, corresponding to ERF, DREB, AP2, RAV and Soloist subfamilies, with the DREB subfamily clustered into four subclades, corresponding to the I, II, III and IV groups and the ERF subfamily clustered into eight subclades, corresponding to V, VI, VII, VIII, IX, X, Xb-L and VI-L groups. The Xb-L and VI-L groups of the ERF subfamily only consisted of *AtERF* genes, suggesting that the ginseng ERF subfamily consisted of only six groups (V–X) (Fig 1). The ML tree (S2 Fig) was essentially the same as the NJ tree, even though the six groups (V–X) of the ERF subfamily were separately distributed in the tree, due to their low bootstrap confidences for both the NJ and ML trees (Fig 1; S2 Fig). When the Arabidopsis *AtERF* genes that were used as the references were examined, they were distributed in every clade and subclade of the *PgERF* gene family tree. Given that Arabidopsis was clustered into a different clade from ginseng in the phylogenetic tree of dicot plant species [35], this result suggested that the *PgERF* gene family is an ancient gene family and originated before ginseng diverged from Arabidopsis.

## Motif identification and multiple sequence alignment

Next, the 93 *PgERF* genes were subjected to conservative analysis for the conserved motifs of their proteins (Fig 2; S3 Fig). A total of 25 conserved motifs were identified for the putative

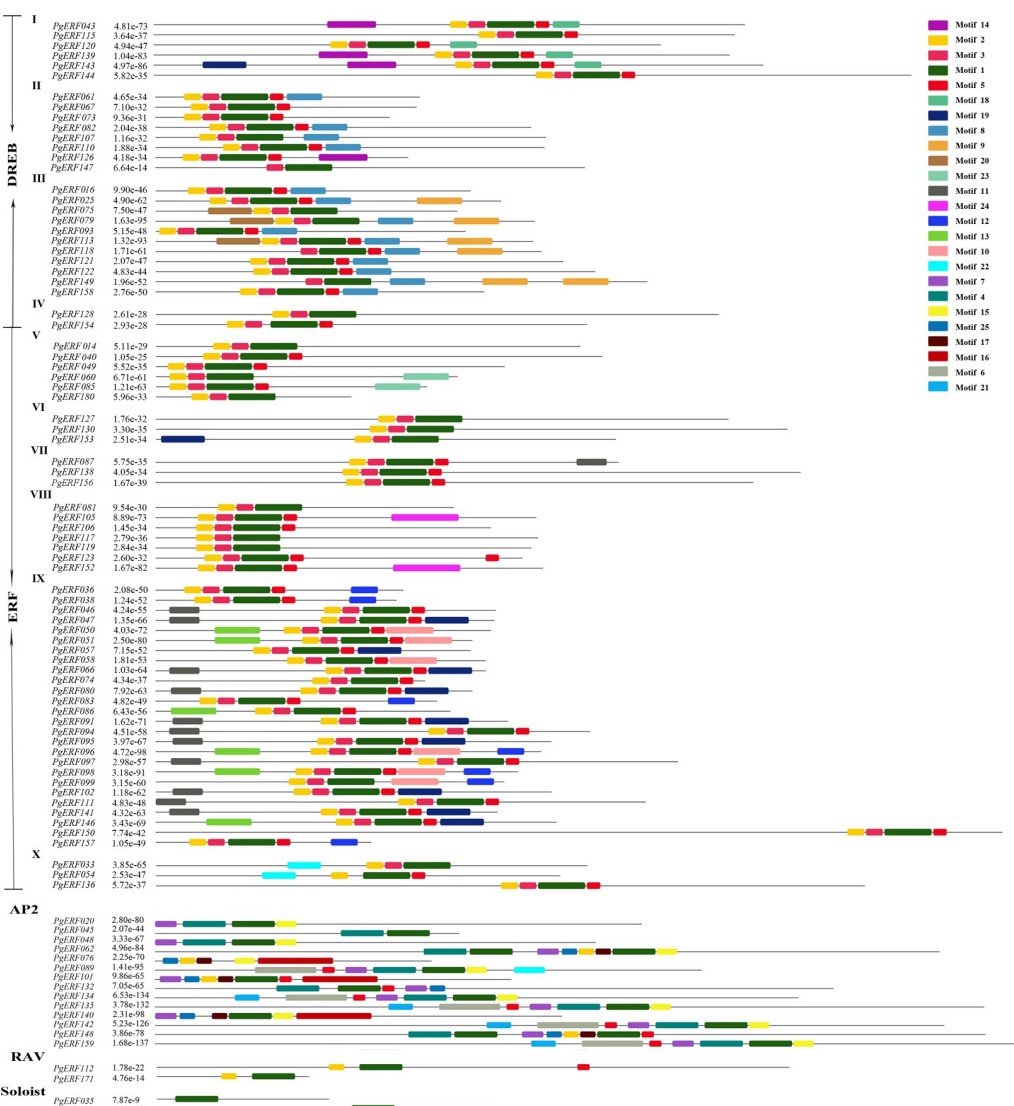

**Fig 2. Distribution of conserved motifs among the gene members of the *PgERF* family.** Each motif is represented by a colored box. Box length corresponds to the motif length.

proteins of the 93 *PgERF* genes and were herein designated as Motif 1 through Motif 25. Motif 1 to Motif 6 were located in the AP2/ERF domain and the remaining 19 motifs, including Motif 7 through Motif 25, were found outside of the AP2/ERF domain. The *PgERF* proteins encoded by gene members of the same subfamily or group contained similar conserved motifs. For example, Motifs 4, 7, 15, 16, 17, 21 and 25 were specifically shared by gene members of the AP2 subfamily. Motifs 14 and 18 were specifically present in the genes of Group I, while Motifs 9 and 20 were only present in the Group III gene members of the ERF subfamily. Motifs 10, 11, 12, 13, and 19 were specific for the DREB subfamily and absent in all four other subfamilies (ERF, AP2, RAV and Soloist subfamilies). These results suggested that most of the 25 motifs were divergent among subfamilies or groups, which might play an important role in their functional divergence [25].

As the featured sequences within the specific domains of transcription factors are critical to their functions [44], the conserved amino acid residues of the AP2 domains were identified for

the genes of both DREB and ERF subfamilies. By aligning the amino acid sequences of the AP2 domains of these two subfamilies in ginseng and Arabidopsis, 14 conserved amino acid residues, including 4G, 6R, 8R, 15W, 16V, 18E, 20R, 22P, 39W, 40L, 49A, 52A, 54D and 72N, and 11 conserved amino acid residues, including 4G, 6R, 8R, 11G, 17I, 30R, 42A, 46Y, 47D, 55G and 63F, were identified for the DREB and ERF subfamilies, respectively (S4 and S5 Figs). Besides, all gene members of the DREB subfamily and ERF subfamily obviously contained the two featured conserved elements, including YRG and RAYD, in their AP2 domains.

## Functional categorization of the *PgERF* genes

To estimate the functional differentiation of the *PgERF* family, all 397 transcripts of the 189 *PgERF* genes identified in this study were annotated and functionally categorized using the Blast2GO software (Version 4.1.9) [45]. Surprisingly, only 195 (49%) of the *PgERF* gene transcripts could be annotated, while the remaining 202 could not be annotated using the database of the Blast2GO software, suggesting the uniqueness of the *PgERF* genes in ginseng. The annotated *PgERF* gene transcripts were categorized into all three primary gene ontology (GO) categories, molecular function (MF), biological process (BP) and cellular component (CC) (Fig 3A). Of the 195 *PgERF* transcripts, 186 (95%) were categorized into all the three primary categories, MF, BP and CC. Only one *PgERF* gene transcript, *PgERF069*, had functions in both BP and CC categories, while two *PgERF* transcripts, *PgERF135-1* and *PgERF135-3*, and six *PgERF* transcripts, *PgERF152-3*, *PgERF152-4*, *PgERF152-1*, *PgERF152-2*, *PgERF105-3* and *PgERF087*, were categorized into BP and CC, respectively. At Level 2, these 195 transcripts were further categorized into eight subcategories, including nucleic acid binding transcription factor activity, binding, metabolic process, cellular process, developmental process, organelle, cell part and cell (Fig 3B). Chi-square test showed that of the eight subcategories, three subcategories, including nucleic acid binding transcription factor activity, binding and cell part subcategories, have been significantly enriched, which is consistent with the transcription regulation functions of the *ERF* genes, while the abundances of the remaining five subcategories are either not changed or significantly reduced relative to the whole genome background control.

Moreover, the *PgERF* transcripts expressed in the roots of 5-, 12-, 18- and 25-year-old plants, 14 tissues of a 4-year-old plant, and the roots of 4-year-old plants of 42 genotypes were further categorized (Fig 4). The *PgERF* transcripts expressed in differently-aged plant roots,

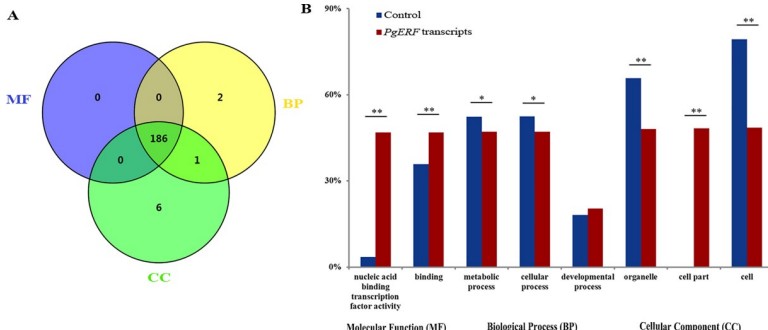

**Fig 3. Functional categorization of the AP2/ERF gene family in ginseng.** (A) Venn diagram of the *PgERF* gene transcripts categorized into three primary categories, biological process (BP), molecular function (MF) and cellular component (CC). (B) The subcategories of the *PgERF* gene transcripts (Level 2). The enrichment of the *PgERF* gene transcripts in each subcategory was calculated using all the gene transcripts of ginseng as the background control. A single asterisk "∗" indicates the significant difference of the number of *PgERF* gene transcripts categorized into the subcategory from that of all the gene transcripts of ginseng at $P \leq 0.05$, while double asterisks "∗∗" indicate the difference at a significance level of $P \leq 0.01$.

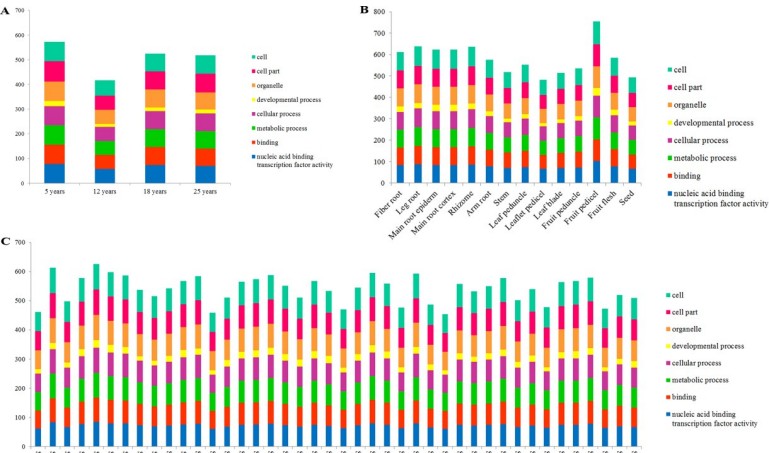

**Fig 4. Variation in functional categories of the *PgERF* gene transcripts.** (A) Variation in functional categories among the roots of differently aged plants. (B) Variation in functional categories among 14 tissues of a 4-year-old plant. (C) Variation in functional categories among the 4-year-old plant roots of 42 genotypes.

different tissues and the roots of different genotypes were all categorized into these eight sub-categories, suggesting that the functions of *PgERF* transcripts were consistent among developmental stages, tissues or genotypes. Nevertheless, a substantial variation of the categorization in the numbers of the *PgERF* transcripts categorized into these eight categories (Level 2) was observed across developmental stages, tissues or genotypes.

## Expression profiles and networks of the *PgERF* genes

To profile the activation patterns of *PgERF* genes, the expressions of all 397 transcripts were quantified in 5-, 12-, 18- and 25-year-old plant roots, 14 4-year-old plant tissues and the roots of 4-year-old plants of 42 genotypes. The expressions of the transcripts varied dramatically across developmental stages, tissues and genotypes, from silenced (0.0 TPM) to 586.3, 666.0 and 1,159.2 TPM, respectively. Of the 397 *PgERF* transcripts, 136 (34.3%), 98 (24.7%) and 83 (20.9%) expressed in all 5-, 12-, 18- and 25-year-old plant roots, all 14 4-year-old plant tissues and the roots of 4-year-old plants of all 42 genotypes, respectively (S5–S7 Tables), while 53 (13.4%), 39 (9.8%) and 14 (3.5%) of the 397 transcripts were development-, tissue- and geno-type-specific, respectively. Nevertheless, the expression of a transcript varied dramatically across developmental stages, tissues and genotypes.

Moreover, we constructed the heatmaps of the *PgERF* genes expressed at different developmental stages of roots, in different tissues, and across different genotypes to find out whether the expressions of the genes were co-regulated. The results showed that although the expression co-regulation was observed for some of the genes at a developmental stage, a single tissue or a genotype and across developmental stages, it was not apparent across tissues or genotypes (Fig 5). For instance, *PgERF140-12*, *PgERF046*, *PgERF089-3*, *PgERF093-3*, *PgERF108-1*, *PgERF184*, *PgERF118-2* and *PgERF170* were apparently co-regulated at a developmental stage and across developmental stages of roots (Fig 5A).

To determine the functional relationships of the *PgERF* genes, the co-expression networks of the *PgERF* transcripts were constructed for 14 tissues of a four-year-old plant and the four-year-old plant roots of 42 genotypes, respectively. Of the 397 *PgERF* gene transcripts, 364 (91.7%) formed a co-expression network ($P \leq 0.05$) in the 14 tissues of the four-year-old plant (S6A Fig). The network consisted of 364 gene transcript nodes, 5,303 co-expression edges and

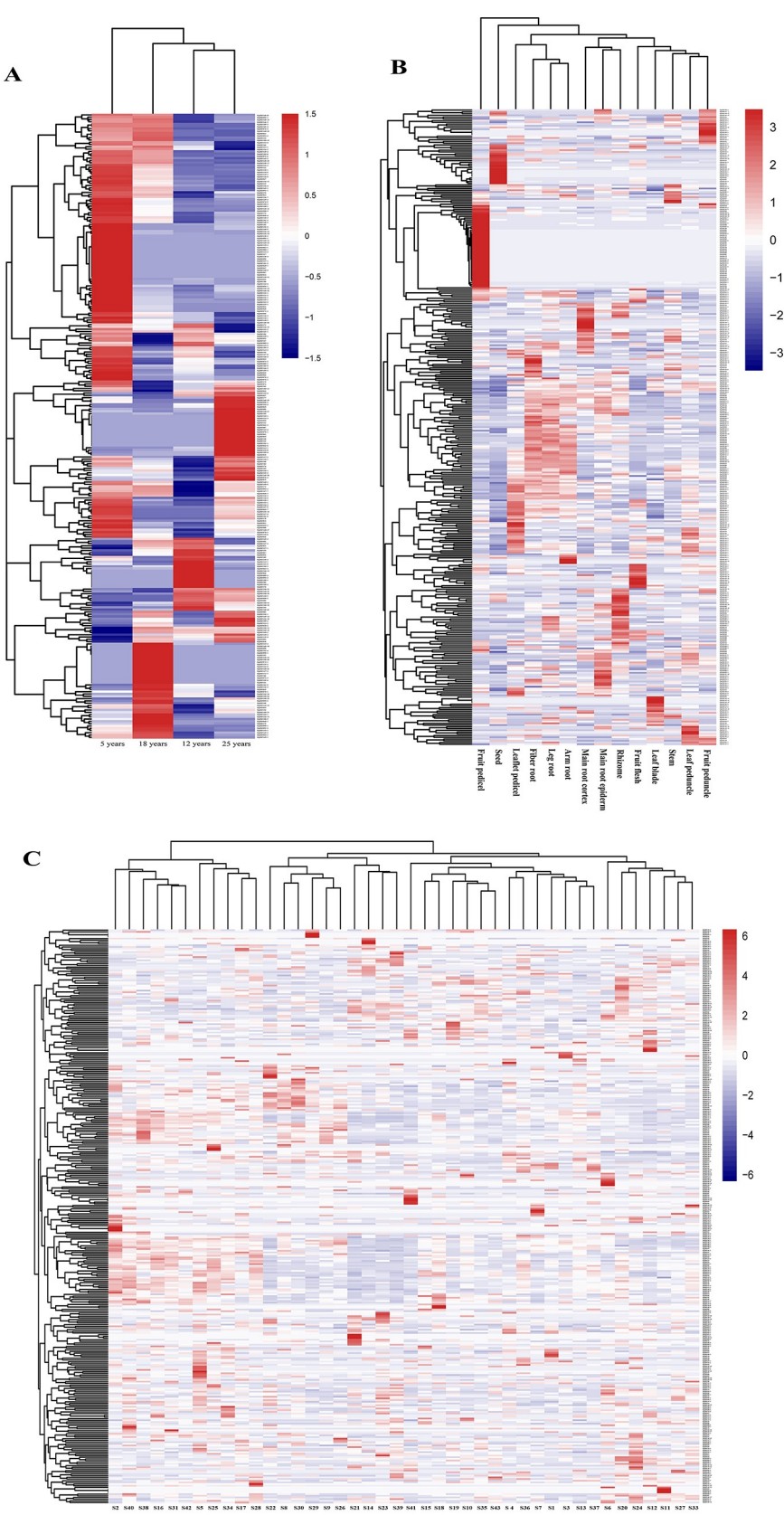

**Fig 5. Expression heatmaps of the *PgERF* gene transcripts at different developmental stages, in different tissues and across genotypes.** (A) In the roots of different aged plants. (B) In the 14 tissues of a 4-year-old plant. (C) In the 4-year-old plant roots of 42 genotypes.

17 closer co-expression clusters (S6A and S6B Fig). Nevertheless, the tendency of this network formation had no substantial difference from that of the network formed from randomly selected ginseng gene transcripts (S6C and S6D Fig). In the four-year-old plant roots of different genotypes, 341 (85.9%) of the 397 *PgERF* gene transcripts formed a co-expression network ($P \leq 0.05$), consisting of 341 gene transcript nodes, 5,606 co-expression edges and 24 clusters (Fig 6A and 6B). The tendency of this network formation was stronger in terms of number of nodes and number of edges than that of the network formed from randomly selected ginseng gene transcripts (Fig 6C and 6D). Together, analysis of these networks revealed that the gene members of the *PgERF* gene family were functionally highly differentiated, even though some of them formed a co-expression network, because the tendency of the network formation was similar to that of randomly-selected unknown genes. This result was consistent with its phylogeny, ancient origin and GO term categorization.

## Expression profiles of the *PgERF* genes in responding to cold stress

As a perennial herb that grows in cold areas, ginseng is frequently suffering from various environmental stresses, especially cold. However, the molecular mechanism underlying the tolerance to cold stress is unclear in ginseng. To test whether the *PgERF* genes are involved in ginseng tolerance to cold stress, the expression patterns of six *PgERF* genes randomly selected from DREB subfamily, i.e., *PgERF073*, *PgERF079*, *PgERF110*, *PgERF115*, *PgERF120* and *PgERF128*, in cold-stressed ginseng hair roots were analyzed by qRT-PCR. *PgERF115* and *PgERF120*, members of Group I of the DREB subfamily, were first up-regulated, but then down-regulated by cold stress (Fig 7D and 7E). The expression of *PgERF115* gradually

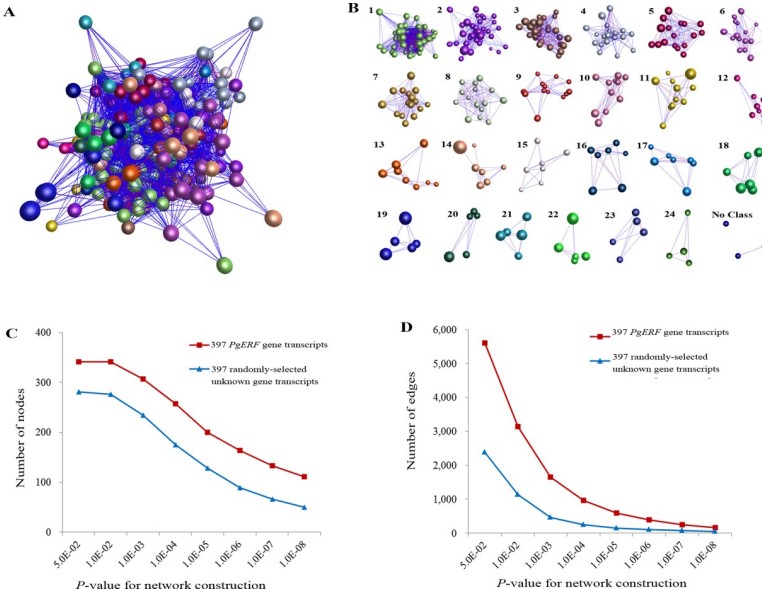

**Fig 6. Co-expression network of the *PgERF* gene transcripts in the 4-year-old plant roots of 42 genotypes.** (A) The co-expression network constructed from 342 of the 397 *PgERF* gene transcripts at $P \leq 0.05$. (B) 17 clusters of the network. (C) Variation in number of nodes in the network of *PgERF* transcripts at different *P*-values. (D) Variation in number of edges in the network of *PgERF* transcripts at different *P*-values.

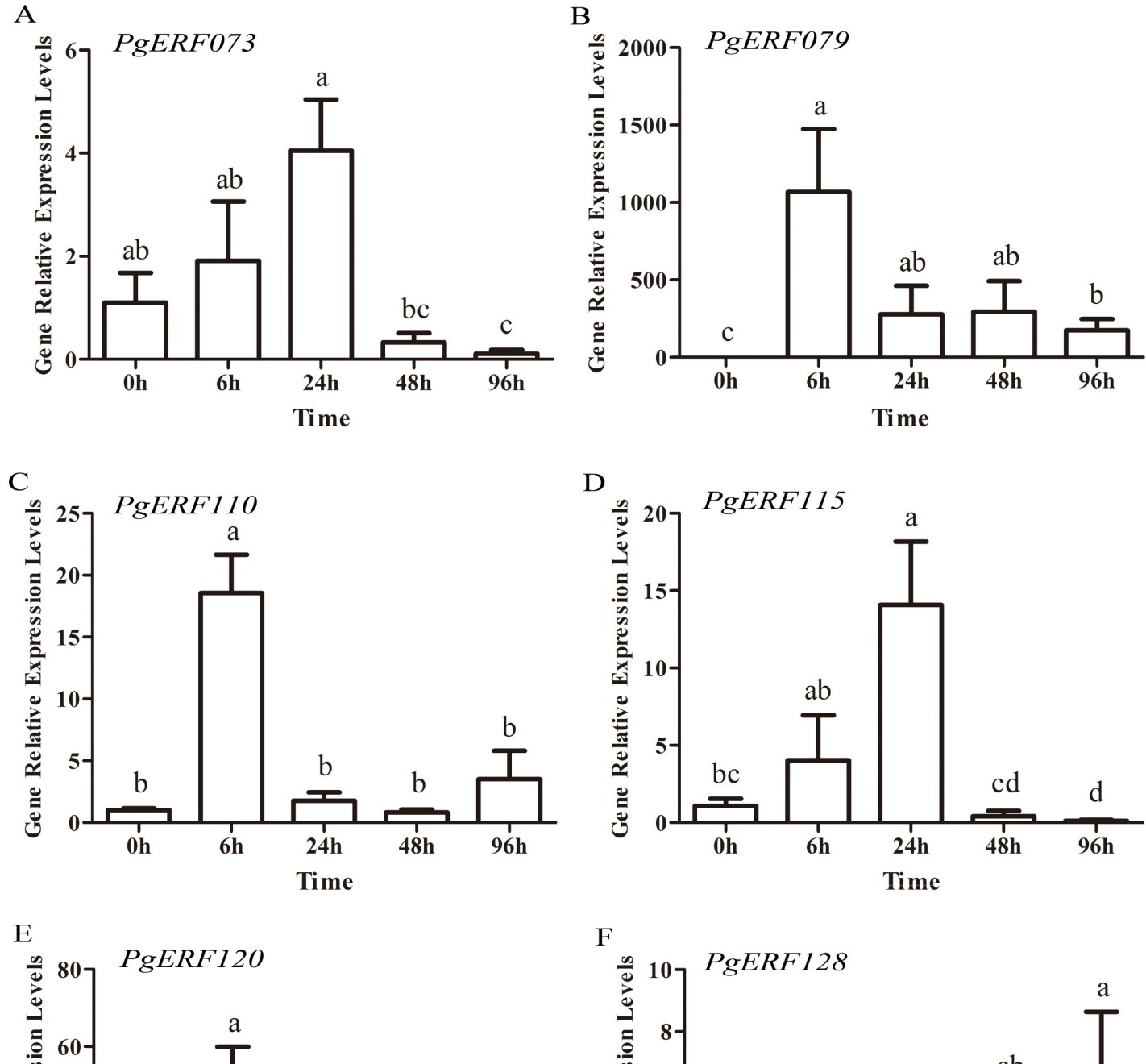

**Fig 7. Expression levels of *PgERF* genes in ginseng hair roots after 0 (control), 6, 24, 48 and 96 h of cold stress treatment.** The values were given as mean ± SD of three biolgical replicates and technical triplicates. Different letters represent significant differences between the treatment means ($P \leq 0.05$, LSD).

increased, relative to the control (0h) as the time of the cold treatment lasted, the expression approached the peak at 24 h (13.1 times higher than the control), and afterward, the expression of *PgERF115* dramatically declined. *PgERF120* responded to cold stress more quickly than *PgERF115*, reaching the expression peak at 6 h after the treatment (26.9 times higher than the control), and then gradually declined to its expression level in the control. Similarly, the expressions of *PgERF073* and *PgERF110*, members of Group II, were also somewhat different from each other. The expression of *PgERF073* showed similar trends to that of *PgERF115*, while the expression of *PgERF110* showed similar trends to *PgERF120* (Fig 7A and 7C). *PgERF079*, a member from Group III of the DREB subfamily, responded rapidly and drastically to the cold stress, whose expression was significantly up-regulated by 1,057.1, 274.4, 290.8 and 173.0 times in the hair roots cold-stressed for 6 h, 24 h, 48 h and 72 h, relative to those of control ($P \leq 0.01$) (Fig 7B). *PgERF128*, which belongs to Group IV of the DREB subfamily, exhibited a gradually increasing trend in expression as the cold treatment time increased from 6 h through 96 h. At 96 h, the expression of *PgERF128* in cold-stressed roots were up-regulated by 5.7 times higher than that in the control hair roots ($P \leq 0.01$).

## Expression profiles of the *PgERF* genes in responding to MeJA

MeJA is a plant hormone and a kind of elicitors that has been widely used in regulation of genes involved in ginsenoside biosynthesis in ginseng [38]. Therefore, we analyzed the expressions of the *PgERF* genes in the ginseng adventitious roots reated with MeJA for 0, 12, 24 and 48 h, respectively. The expressions of the *PgERF* gene transcripts in the control and MeJA-treated adventitious roots varied from silent (0 TPM) to 197.7 TPM (S8 Table). Of the 397 *PgERF* gene transcripts profiled, 173 (43.6%) expressed and 109 (27.5%) silenced in the control and all treated adventitious roots, and the remaining 115 (29.0%) either expressed or silenced in these adventitious roots. The expressions of the 288 *PgERF* gene transcripts expressed in the adventitious roots were visualized by the expression heatmap (Fig 8). Overall, all the 288 *PgERF* gene transcripts responded to the MeJA treatment, with 136 of them up-regulated and 152 down-regulated by MeJA. Among the three treatment times, 12 h, 24 h and 48 h, the responses of these *PgERF* gene transcripts to MeJA varied from time to time.

## Discussion

The AP2/ERF gene family has been studied in several plant species of economical or biological importance at the whole genome and transcriptome level, due to its important roles in various biological processes, including growth and development, and responses to environmental stresses [5]. These species include Arabidopsis [6], rice [7], wheat [46], maize [26], cotton [3], grapevine [28], cucumber [47] and rubber tree [48]. This study has comprehensively investigated the AP2/ERF genes in ginseng using several ginseng transcriptome and genome databases. The *PgERF* gene family is a large gene family, consisting of 189 or more gene members. This family size is in consistence with those of the AP2/ERF gene families identified in Arabidopsis [6], rice [7] and grapevine [28]. Although the *PgERF* family size quite differs from those identified in wheat [46], maize [26], cotton [3], cucumber [47] and rubber tree [48], due to the variation of gene family size within a plant species [49] and the difference of the databases used for these analyses, the *PgERF* gene family is unambiguously classified into five subfamilies, ERF, DREB, AP2, RAV and Soloist, as those identified in Arabidopsis [6], rice [7] and

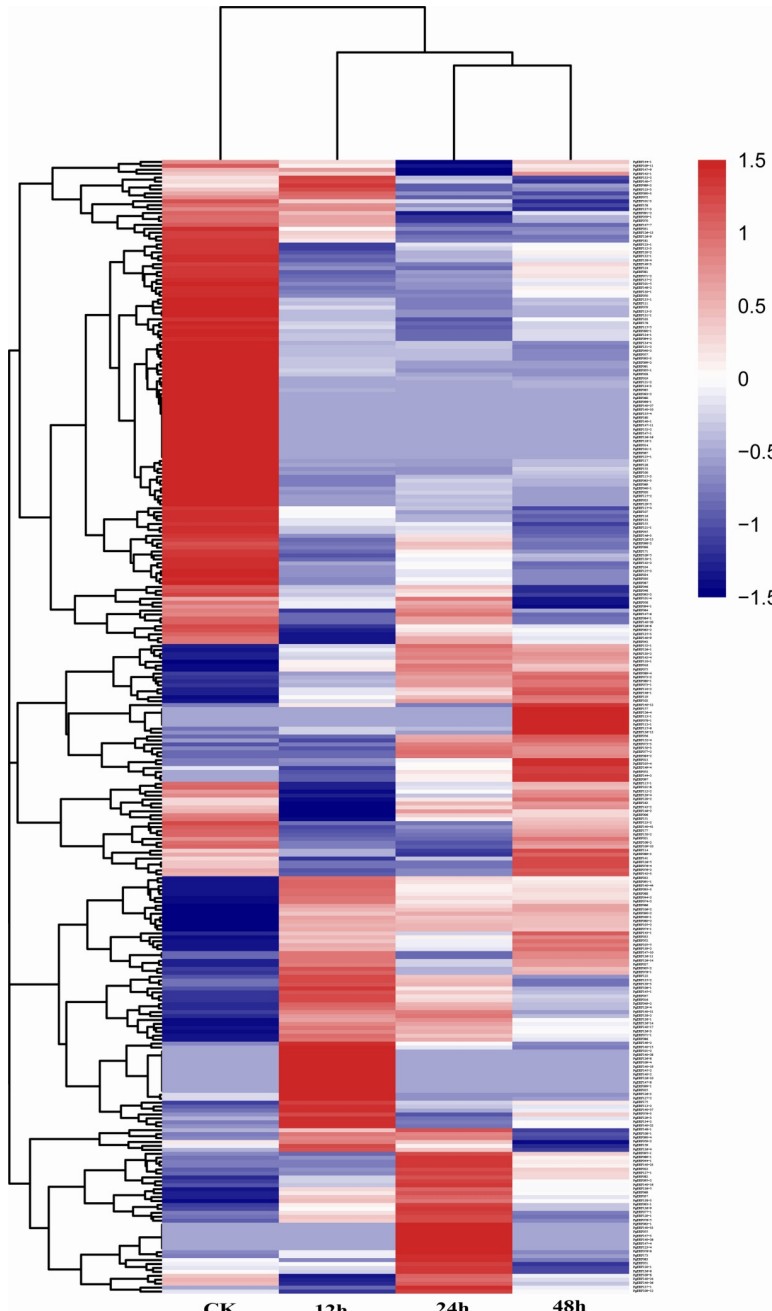

**Fig 8. Expression heatmap of the *PgERF* gene transcripts treated with 200 μM MeJA for 0, 12, 24 and 48 h, respectively.**

grapevine [28]. These results indicate that the *PgERF* gene family has a similar functional differentiation pattern to those in these three species.

It has been the consensus that the conserved motifs of the AP2/ERF transcription factors are crucial to their functions, such as nuclear localization and transcriptional activity [7]. The DNA binding domain of AP2/ERF transcription factors, the AP2 domain, is highly conserved in plant species [50,51]. The AP2 domain of the *PgERF* genes is also highly conserved. This study has identified 14 and 11 completely conserved amino acid residues through the gene

members of the DREB and ERF subfamilies, respectively, in both ginseng and Arabidopsis (S3 and S4 Figs). In Arabidopsis, the two conservative elements, YRG and RAYD, were shown to be critical to the binding of the AP2/ERF transcription factors to the promoter regions of the target genes and the modulating of their expression [29]. The conservative YRG and RAYD elements identified in the AP2 domain of the DREB and ERF subfamilies in ginseng may suggest their necessity for similar functions of the *PgERF* genes. Nevertheless, subtle variation exists among the amino acid sequences of the *PgERF* transcription factors, which has led to the separation of the DREB subfamily from the ERF subfamily. The difference between the DREB and ERF subfamilies might result in their functional divergence in ginseng. Moreover, the "EIR" in the AP2 domain was found to be shared by all gene members of the ERF subfamily and the vast majority of the gene members of the DREB subfamily in both ginseng and Arabidopsis, while the "EVR" exists only in Group III of the DREB subfamily in both ginseng and Arabidopsis and only in Group II (*PgERF061*) of the DREB subfamily in ginseng. It has been reported that the sequence similarity of the conserved motifs present outside of the DNA binding domain was low [6,27,43]. In ginseng, 19 conserved motifs, except for Motif 1 to Motif 6, were identified outside of the AP2 domain. The vast majority of these 19 motifs were found to be divergent across subfamilies or even subfamily groups in ginseng. The subfamily/group-specific distribution pattern of these motifs might have led to the functional divergence between subfamilies or groups of the *PgERF* transcription factors.

Because different transcripts alternatively spliced from the same gene may have different functions [52], this study has annotated and functionally categorized the 397 *PgERF* transcripts spliced from the 189 *PgERF* genes. The *PgERF* transcripts were categorized into eight subcategories at Level 2. This result suggests a substantial functional differentiation of the *PgERF* genes; however, the differentiation is much smaller than those observed in the *PgNBS* gene family [35], the *PgRLK* gene family [53] and the *PgCYP* gene family [54] in ginseng. Interestingly, only two of the eight subcategories, especially subcategory of nucleic acid binding transcription factor activity, were significantly up-enriched. This result is consistent with the roles of the *PgERF* genes as transcription factors by binding to the promoters of target genes. While the functions of the AP2/ERF genes have been shown to play important roles in plant growth and development, response to stresses and signal pathway in the model plant species, such as Arabidopsis [5] and rice [55], further research is needed to determine the functions of the *PgERF* genes in ginseng.

Companioned with their functional differentiation, the expressions of 397 *PgERF* transcripts dramatically varied in a tissue, at a development stage or in a genotype. Moreover, the type and number of expressed *PgERF* transcripts also diversified tempo-spatially and across genotypes. The differential expressions of the AP2/ERF genes were previously reported in other plant species, but mainly among tissues [25,56]. Furthermore, the numbers of the *PgERF* transcripts categorized into each subcategory varied across tissues, developmental stages or genotypes. These variations might be an indication of their functional differentiation. On the other hand, co-expression network analysis revealed that most (>86%) of these *PgERF* transcripts express correlatively and tend to form a co-expression network in different tissues or different genotypes. These results suggest that the *PgERF* genes have functionally differentiated, but they still maintain somehow functionally collaborative.

As a perennial herb, ginseng frequently suffers from different environmental stresses. It was reported that the members of AP2/ERF family, especially those of the DREB subfamily, played an essential role in response to biotic and abiotic stresses [16,17,57]. To explore the potential of the AP2/ERF genes of DREB subfamily in response to cold stress in ginseng, the expressions of six genes randomly selected from the DREB subfamily were analyzed under cold stress using qRT-PCR. The expression of *PgERF079*, from Group III or A1 group of the DREB

subfamily, was changed by up to 1,057 times, relative to the control, confirming it played an extremely important role in response to cold stress. A-1 group was considered to be a major regulator of cold-stress responses, as overexpressing any of the three cold-inducible DREB1s, *DREB1A/CBF3* (*AT4G25480*), *DREB1B/CBF1* (*AT4G25490*) and *DREB1C/CBF2* (*AT4G25470*), significantly improved cold tolerance in Arabidopsis [57–59]. This study also shows that the expression levels of the remaining five genes from other groups of the DREB subfamily, i.e., *PgERF073*, *PgERF110*, *PgERF115*, *PgERF120* and *PgERF128*, significantly changed, when the ginseng hair roots were subjected to cold stress. Therefore, we speculate that in addition to A1 group, the genes from the other groups of the DREB subfamily may also play a role in response to cold stress in ginseng. The results of these cold-inducible genes provide information useful for the functional studies of *PgERF* genes in ginseng.

It has been reported that some genes of the AP2/ERF family are involved in response to hormone signals in plants [60–62]. It was showed that MeJA, as a signaling molecule, was rapidly synthesized in plants, when subjected to various biotic and abiotic stresses, and then, induced defense-related responses to the stresses and regulate plant growth and development [63]. MeJA has been also used as an effective elicitor, since it can stimulate the biosynthesis of plant secondary metabolites [64–66]. The biosynthesis and accumulation of ginsenosides, a cluster of important secondary metabolites and the most valuable bioactive components in ginseng, were reported to be induced by MeJA [67,68]. This study revealed that addition of exogenous MeJA to adventitious roots dramatically changed the expression of a majority of the *PgERF* gene transcripts. The expressions of some of the transcripts were up-regulated, while those of others were down-regulated or inhibited by MeJA, relative to the control not treated by MeJA. Given the demonstrated functions of MeJA in plant responses to biotic and abiotic stresses, growth and development and secondary metabolite biosynthesis in other plant species [57,60–66], the *PgERF* genes may also be involved in these processes, including the biosynthesis of ginsenosides.

## Conclusions

The present study has, for the first time, identified and systematically characterized the AP2/ERF family in ginseng, i.e., the *PgERF* gene family. A total of 189 *PgERF* genes that actively expressed in 14 tissues of a four-year-old ginseng plant were identified, from which 397 transcripts were alternatively spliced. These *PgERF* genes were classified into five subfamilies, DREB, ERF, AP2, RAV and Soloist, as those previously identified in Arabidopsis. As expected, the conserved motifs that characterize the AP2/ERF family and several conserved domains were identified among the members of the *PgERF* gene family. Nevertheless, the transcripts of the *PgERF* genes were apparently categorized into eight subcategories by GO, indicating their functional differentiation. Along with their functional differentiation, the expressions of the *PgERF* genes, including the type, number and expression level of their transcripts, have also substantially diversified tempo-spatially and across genotypes. In spite of these differentiations, most of the *PgERF* genes remain to co-express and form a co-expression network, suggesting that most of the genes in the *PgERF* gene family remain functionally correlated. These *PgERF* genes and findings provide resources and knowledge necessary for family-wide functional analysis of the *PgERF* genes and determination of their roles in plant responses to biotic and abiotic stresses, growth and development, and biosynthesis of secondary metabolites, especially ginsenosides, in *P. ginseng* and related species.

## Supporting information

**S1 Fig.** The Venn diagram of the sequence comparison between 397 *PgERF* transcripts identified in this study (left) and 342 *AP2/ERF* TFs CDS downloaded from Ginseng Genome

Database (right).
(PDF)

**S2 Fig. Phylogenetic tree of the *AP2/ERF* gene family present in ginseng and Arabidopsis.**
(PDF)

**S3 Fig. Conserved motifs identified from the *PgERF* genes.**
(PDF)

**S4 Fig. Comparison of the amino acid sequences of the protein AP2 domains of the DREB subfamily between ginseng and Arabidopsis.**
(PDF)

**S5 Fig. Comparison of the amino acid sequences of the AP2 domains of the ERF subfamily proteins from ginseng and Arabidopsis.**
(PDF)

**S6 Fig. Co-expression network of the *PgERF* gene transcripts in 14 tissues of a 4-year-old ginseng plant.**
(PDF)

**S1 Table. List of primers used for qRT-PCR amplification of selected genes.**
(XLSX)

**S2 Table. The nucleic acid sequences of the 397 predicted *PgERF* gene transcripts.**
(XLSX)

**S3 Table. Sequence comparison between 342 *AP2/ERF* TFs CDS downloaded from Ginseng Genome Database and 397 *PgERF* transcripts identified in this study.**
(XLSX)

**S4 Table. The amino acid sequences and properties of putative proteins encoded by the 221 predicted *PgERF* gene transcripts that had complete AP2 domains within ORFs.**
(XLSX)

**S5 Table. Expression profiles of the *PgERF* gene transcripts in the roots of 5-, 12-, 18- and 25-year-old ginseng plants.**
(XLSX)

**S6 Table. Expression profiles of the *PgERF* gene transcripts in 14 tissues of a 4-year-old ginseng plant.**
(XLSX)

**S7 Table. Expression profiles of the *PgERF* gene transcripts in the four-year-old plant roots of 42 genotypes representing the diversity of Jilin ginseng in Northeastern China.**
(XLSX)

**S8 Table. Expression profiles of *PgERF* gene transcripts in the adventitious roots of ginseng treated with 200 μM MeJA for 0, 12, 24 and 48 h, with three replications.**
(XLSX)

## Author Contributions

**Data curation:** Jing Chen, Yuanhang Zhou, Qi Zhang.

**Funding acquisition:** Meiping Zhang, Yi Wang.

**Resources:** Yi Wang.

**Software:** Jing Chen, Qian Liu, Li Li, Chunyu Sun, Kangyu Wang, Yanfang Wang, Mingzhu Zhao, Hongjie Li, Yilai Han, Ping Chen, Ruiqi Li, Jun Lei, Meiping Zhang.

**Visualization:** Yuanhang Zhou.

**Writing – original draft:** Jing Chen.

**Writing – review & editing:** Meiping Zhang.

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
