## [Decision Letter · Decision Letter 0]

17 Dec 2019

PONE-D-19-31960

Structural variation, functional differentiation and expression characteristics of the AP2/ERF gene family and its response to cold stress and methyl jasmonate in Panax ginseng C.A. Meyer

PLOS ONE

Dear Mr Wang,

Thank you for submitting your manuscript to PLOS ONE. After careful consideration, we feel that it has merit but does not fully meet PLOS ONE’s publication criteria as it currently stands. Therefore, we invite you to submit a revised version of the manuscript that addresses the points raised during the review process.

We would appreciate receiving your revised manuscript by Jan 31 2020 11:59PM. To enhance the reproducibility of your results, we recommend that if applicable you deposit your laboratory protocols in protocols.io, where a protocol can be assigned its own identifier (DOI) such that it can be cited independently in the future. For instructions see: http://journals.plos.org/plosone/s/submission-guidelines#loc-laboratory-protocols

We look forward to receiving your revised manuscript.

Kind regards,

Anil Kumar Singh, Ph.D.

Academic Editor

PLOS ONE

Journal Requirements:

2) PLOS requires an ORCID iD for the corresponding author in Editorial Manager on papers submitted after December 6th, 2016. Please ensure that you have an ORCID iD and that it is validated in Editorial Manager. To do this, go to ‘Update my Information’ (in the upper left-hand corner of the main menu), and click on the Fetch/Validate link next to the ORCID field. This will take you to the ORCID site and allow you to create a new iD or authenticate a pre-existing iD in Editorial Manager. Please see the following video for instructions on linking an ORCID iD to your Editorial Manager account: https://www.youtube.com/watch?v=_xcclfuvtxQ

3) Please include captions for your Supporting Information files at the end of your manuscript, and update any in-text citations to match accordingly. Please see our Supporting Information guidelines for more information: http://journals.plos.org/plosone/s/supporting-information.

Reviewers' comments:

Reviewer's Responses to Questions

**Comments to the Author**

1. Is the manuscript technically sound, and do the data support the conclusions?

Reviewer #1: Yes

Reviewer #2: Yes

2. Has the statistical analysis been performed appropriately and rigorously? 

Reviewer #1: I Don't Know

Reviewer #2: Yes

3. Have the authors made all data underlying the findings in their manuscript fully available?

Reviewer #1: Yes

Reviewer #2: Yes

4. Is the manuscript presented in an intelligible fashion and written in standard English?

Reviewer #1: Yes

Reviewer #2: Yes

5. Review Comments to the Author

Reviewer #1: The manuscript submitted by Chen et al., and titled “Structural variation, functional differentiation and expression characteristics of the AP2/ERF gene family and its response to cold stress and methyl jasmonate in Panax ginseng C.A. Meyer” addresses a good research topic and provides a very comprehensive review of the identification and characterization of the AP2/ERF family present in ginseng. Authors have classified PgERFs on the basis of structural similarity and presence of AP2/ERF domains into five sub families. Authors have introduced a series of well-designed experiments including phylogenetic analysis, motif identification and multiple sequence alignment for characterization of PgERF genes. They have carried out functional characterization and expression profiling of these genes using different transcriptomic databases. Authors have shown the response of PgERF genes to cold stress, suggesting their role in freezing tolerance of ginseng. They have also shown the response of PgERF genes to MeJA, suggesting their role in growth and development and secondary metabolite biosynthesis. The work would contribute towards the functional validation of the PgERF genes in responses to biotic and abiotic stresses, growth and development, and biosynthesis of ginsenosides, in Panax ginseng and related species.

In general, the research hypotheses and materials and methods used in this research are scientifically sound; the results are well represented, and the discussion of results are well balanced. The manuscript can be accepted with minor revisions.

Minor concerns:

1- Introduction part:

Page 4, Line no. 80: Arabidopsis thaliana

Page 4, Line no. 82: remove “and” before grapevine

Page 4, Line no. 83: Change “Ginseng” to “Ginseng”

2- The materials and methods section is very well-written, adequately detailed and well-organized. However, authors

should mention the method used to describe the statistical significance of the data represented (p≤0.05 or p≤0.01).

3- The Results section is well-written. However, there are some errors:-

Page 10, Line no. 206: Change “Furthermore” to “Moreover”

Page 10, Line no. 207: Either mention “A1 to A6 and B1 to B6” in Figure 1 or remove these mentions from the text

in manuscript.

Page 10, Line no. 214: remove “also”

Page 11, Line no. 223: change “(Fig.2)” to “(Fig.1)”

In Figure 2, group VII and group VIII are not mentioned. (labeling error).

Page 13, Line no. 275: Change “Furthermore” to “Moreover”

Label the Figure 7 with A, B, C, D, E, F as you mentioned in text line no’s. 334, 342, 345 and 347.

4- The discussion section is well written, few errors are:

Page 18, Line no. 383: change “the” to “that”

Page 20, Line no. 435, line no. 437, line no. 443: remove “A1 group” or mention it in Figure1.

5- The reference section contain many typos and lack of punctuation marks such as:-

Page 25, Line no. 534: Arabidopsis , journal volume and page no’s missing.

Page 26, Line no. 566: BMC Genomics

Page 27, Line no. 590: journal name should be in Italic

Page 27, Line no. 591: remove full stop after (2018)

Page 28, Line no. 605: Journal name should be in Italic, change colon with comma

Page 29, Line no. 627: Use reference style suggested by Plos One journal

Page 30, Line no. 642: Arabidopsis ; Change colon with comma

Page 30, Line no. 645: Change colon with comma

Page 30, Line no. 654: Journal volume and page no’s missing.

Reviewer #2: Chen et al., attempted to investigate the structural variation, functional diversity and expression profile of AP2/ERF gene family using transcriptomic dataset in ginseng. They claimed a total of 189 putative AP2/ERF genes and 397 PgERF gene transcripts diversified across different tissues and demonstrated the expression patterns in response to cold resistance and Methyl jasmonate (MeJA) treatment. The authors have been provided lot of data, facts and information. However, the manuscript lacks the detailed interpretations that should be addressed by authors in the manuscript.

Here are few queries for the manuscript:

What issue authors want to address in the manuscript by mentioning the clustering in so and so group? Authors should highlight the importance in evolution and explain the importance of phylogeny of AP2/ERF family for its evolution.

Why authors used only NJ method? And what happens if they used ML method to divulge the phylogeny?

Why they used TPM value not RPKM?? And which is better to consider during analysis of expression profile?

Authors should mention the retrieval or access date of downloaded dataset in materials and method section

Authors have well written the manuscript. However, there is still a space for improvement while writing in English. For eg line 25-26 (A total… PgERF189), line 46…48 (Therefore,… the stresses)

6. PLOS authors have the option to publish the peer review history of their article (what does this mean?). If published, this will include your full peer review and any attached files.

Reviewer #1: Yes: VISHAL SHARMA

Reviewer #2: No

---

## [Author Response · Author response to Decision Letter 0]

27 Jan 2020

Dear reviewers,

Thank you for your comments concerning our manuscript entitled “Structural variation, functional differentiation and expression characteristics of the AP2/ERF gene family and its response to cold stress and methyl jasmonate in Panax ginseng C.A. Meyer” (Manuscript Number: PONE-D-19-31960). These comments are constructive, valuable and very helpful for improving our manuscript. We have studied them very carefully and revised the manuscript as suggested. The revisions are indicated in the file: “Track Changes._Manuscript_Rev1_Chen et al.docx”, and the point to point responses are summarized as follows:

Reviewer 1:

Reviewer #1 Suggestions:

Reviewer #1: The manuscript submitted by Chen et al., and titled “Structural variation, functional differentiation and expression characteristics of the AP2/ERF gene family and its response to cold stress and methyl jasmonate in Panax ginseng C.A. Meyer” addresses a good research topic and provides a very comprehensive review of the identification and characterization of the AP2/ERF family present in ginseng. Authors have classified PgERFs on the basis of structural similarity and presence of AP2/ERF domains into five sub families. Authors have introduced a series of well-designed experiments including phylogenetic analysis, motif identification and multiple sequence alignment for characterization of PgERF genes. They have carried out functional characterization and expression profiling of these genes using different transcriptomic databases. Authors have shown the response of PgERF genes to cold stress, suggesting their role in freezing tolerance of ginseng. They have also shown the response of PgERF genes to MeJA, suggesting their role in growth and development and secondary metabolite biosynthesis. The work would contribute towards the functional validation of the PgERF genes in responses to biotic and abiotic stresses, growth and development, and biosynthesis of ginsenosides, in Panax ginseng and related species.

In general, the research hypotheses and materials and methods used in this research are scientifically sound; the results are well represented, and the discussion of results are well balanced. The manuscript can be accepted with minor revisions.

Authors’ Response:

Thanks.

Minor concerns:

Reviewer #1 Suggestions:

1- Introduction part:

Page 4, Line no. 80: Arabidopsis thaliana

Page 4, Line no. 82: remove “and” before grapevine

Page 4, Line no. 83: Change “Ginseng” to “Ginseng”

Authors’ Response:

Thanks for carefully reading and editing. We are sorry for the inaccurate writing. As your suggesting, we have corrected the wrong writing and checked throughout the manuscript.

Reviewer #1 Suggestions:

2- The materials and methods section is very well-written, adequately detailed and well-organized. However, authors should mention the method used to describe the statistical significance of the data represented (p≤0.05 or p≤0.01).

Authors’ Response:

Thanks. We have added the statistical methods used to describe the statistical significance of the data represented (p≤0.05 or p≤0.01), where appropriate.

Reviewer #1 Suggestions:

3- The Results section is well-written. However, there are some errors:-

Page 10, Line no. 206: Change “Furthermore” to “Moreover”

Page 10, Line no. 207: Either mention “A1 to A6 and B1 to B6” in Figure 1 or remove these mentions from the text in manuscript.

Page 10, Line no. 214: remove “also”

Page 11, Line no. 223: change “(Fig.2)” to “(Fig.1)”

In Figure 2, group VII and group VIII are not mentioned. (labeling error).

Page 13, Line no. 275: Change “Furthermore” to “Moreover”

Label the Figure 7 with A, B, C, D, E, F as you mentioned in text line no’s. 334, 342, 345 and 347.

Authors’ Response:

Thanks for carefully reading and editing. As your suggesting, we have corrected the wrong writing in the text and the mistakes in Figure 2 and Figure 7. Besides, we added the mention of ‘A1 to A6 and B1 to B6’ in the Figure 1.

Reviewer #1 Suggestions:

4- The discussion section is well written, few errors are:

Page 18, Line no. 383: change “the” to “that”

Page 20, Line no. 435, line no. 437, line no. 443: remove “A1 group” or mention it in Figure1.

Authors’ Response:

Thanks for carefully reading and editing. As your suggesting, we have corrected the wrong writing in the text added the mention of ‘A1 to A6 and B1 to B6’ in the Figure 1.

Reviewer #1 Suggestions:

5- The reference section contain many typos and lack of punctuation marks such as:-

Page 25, Line no. 534: Arabidopsis, journal volume and page no’s missing.

Page 26, Line no. 566: BMC Genomics

Page 27, Line no. 590: journal name should be in Italic

Page 27, Line no. 591: remove full stop after (2018)

Page 28, Line no. 605: Journal name should be in Italic, change colon with comma

Page 29, Line no. 627: Use reference style suggested by Plos One journal

Page 30, Line no. 642: Arabidopsis ; Change colon with comma

Page 30, Line no. 645: Change colon with comma

Page 30, Line no. 654: Journal volume and page no’s missing.

Authors’ Response:

Thanks for carefully reading and editing. We are sorry for the inaccurate writing and mistakes in the reference section. As your suggesting, we have added the missing data of journal, corrected the wrong style. Besides, we have checked throughout the reference section.

Reviewer #2 Suggestions:

Reviewer #2: Chen et al., attempted to investigate the structural variation, functional diversity and expression profile of AP2/ERF gene family using transcriptomic dataset in ginseng. They claimed a total of 189 putative AP2/ERF genes and 397 PgERF gene transcripts diversified across different tissues and demonstrated the expression patterns in response to cold resistance and Methyl jasmonate (MeJA) treatment. The authors have been provided lot of data, facts and information. However, the manuscript lacks the detailed interpretations that should be addressed by authors in the manuscript.

Authors’ Response:

Thanks. We have added the detailed interpretations where appropriate (also see below).

Here are few queries for the manuscript:

Reviewer #2 Suggestions:

What issue authors want to address in the manuscript by mentioning the clustering in so and so group? Authors should highlight the importance in evolution and explain the importance of phylogeny of AP2/ERF family for its evolution.

Authors’ Response:

Thanks. It is a very good point. We have highlighted and explained the importance of the phylogeny of AP2/ERF family for the family evolution, and inferred the origin and evolution of the PgERF family, based on its phylogenetic tree in the section of Phylogenetic analysis of the PgERF gene family, Results. 

Reviewer #2 Suggestions:

Why authors used only NJ method? And what happens if they used ML method to divulge the phylogeny?

Authors’ Response:

Thanks. We have also constructed the phylogenetic tree using the ML method. The resultant ML tree is shown in Fig. S2. The ML tree is essentially the same as the NJ tree. We have presented the comparison between the two trees in the section of Phylogenetic analysis of the PgERF gene family, Results. 

Reviewer #2 Suggestions:

Why they used TPM value not RPKM?? And which is better to consider during analysis of expression profile?

Authors’ Response:

Thanks. It is a very good point. Three methods, TPM (Transcripts Per Million or Transcripts Per Kilobase Million), RPKM (Reads Per Kilobase Million) and FPKM (Fragments Per Kilobase Million), are used to present the expression of a transcript or gene quantified by RNA-seq. All three methods normalize the expressions of transcripts or genes that have different lengths among samples that have different sequencing depths with sequencing depth and transcript or gene length, but the normalization order of them are different for three methods. TPM normalizes the transcript or gene length first and then the sequencing depth, while RPKM and FKPM normalize the sequencing depth first and then the transcript or gene length. Therefore, TPM is more desirable for comparative analysis between samples, such as those in this study, than RPKM and FKPM. This is because TPM always has the same denominator, regardless of what samples you are looking at, while the denominator of RPKM or FKPM could be different for different samples, thus being not comparable. 

For more information, please see https://www.rna-seqblog.com/rpkm-fpkm-and-tpm-clearly-explained/

Reviewer #2 Suggestions:

Authors should mention the retrieval or access date of downloaded dataset in materials and method section

Authors’ Response:

Thanks. We have added a sentence to indicate where and how the datasets used for this study are retrieved in the materials and method section.

Reviewer #2 Suggestions:

Authors have well written the manuscript. However, there is still a space for improvement while writing in English. For eg line 25-26 (A total… PgERF189), line 46…48 (Therefore,… the stresses)

Authors’ Response:

Thanks. Both we and an American professor of genomics have carefully read and edited the manuscript, with a focus on English, including the correction of the sentences indicated above by the reviewer.

---

## [Decision Letter · Decision Letter 1]

27 Feb 2020

Structural variation, functional differentiation and expression characteristics of the AP2/ERF gene family and its response to cold stress and methyl jasmonate in Panax ginseng C.A. Meyer

PONE-D-19-31960R1

Dear Dr. Wang,

We are pleased to inform you that your manuscript has been judged scientifically suitable for publication and will be formally accepted for publication once it complies with all outstanding technical requirements.

With kind regards,

Anil Kumar Singh, Ph.D.

Academic Editor

PLOS ONE

Additional Editor Comments (optional):

Reviewers' comments:

Reviewer's Responses to Questions

**Comments to the Author**

1. If the authors have adequately addressed your comments raised in a previous round of review and you feel that this manuscript is now acceptable for publication, you may indicate that here to bypass the “Comments to the Author” section, enter your conflict of interest statement in the “Confidential to Editor” section, and submit your "Accept" recommendation.

Reviewer #1: All comments have been addressed

Reviewer #2: All comments have been addressed

2. Is the manuscript technically sound, and do the data support the conclusions?

Reviewer #1: Yes

Reviewer #2: Yes

3. Has the statistical analysis been performed appropriately and rigorously? 

Reviewer #1: Yes

Reviewer #2: Yes

4. Have the authors made all data underlying the findings in their manuscript fully available?

Reviewer #1: Yes

Reviewer #2: Yes

5. Is the manuscript presented in an intelligible fashion and written in standard English?

Reviewer #1: Yes

Reviewer #2: Yes

6. Review Comments to the Author

Reviewer #1: Authors have added the statistical methods used to describe the statistical significance of the data represented. The research hypotheses and materials and methods used in this research are scientifically sound.

Authors have also made corrections to the figure 2 and figure 7. The results are well represented, and the discussion of results are well balanced now.

Authors have adequately addressed the comments raised in a previous round of review. I feel that this manuscript is now acceptable for publication.

Reviewer #2: (No Response)

7. PLOS authors have the option to publish the peer review history of their article (what does this mean?). If published, this will include your full peer review and any attached files.

Reviewer #1: Yes: VISHAL SHARMA

Reviewer #2: No

---

## [Editor Report · Acceptance letter]

3 Mar 2020

PONE-D-19-31960R1 

Structural variation, functional differentiation and expression characteristics of the AP2/ERF gene family and its response to cold stress and methyl jasmonate in *Panax ginseng* C.A. Meyer 

Dear Dr. Wang:

I am pleased to inform you that your manuscript has been deemed suitable for publication in PLOS ONE. Congratulations! Your manuscript is now with our production department. 

With kind regards,

on behalf of

Dr. Anil Kumar Singh 

Academic Editor

PLOS ONE